

# The effect of recent competition between the native *Anolis oculatus* and the invasive *A. cristatellus* on display behavior

Claire M. S. Dufour[1], Anthony Herrel[2] and Jonathan B. Losos[1,3]

[1] Department of Organismic and Evolutionary Biology, Museum of Comparative Zoology, Harvard University, Cambridge, MA, USA
[2] Département 'Adaptations du vivant', UMR 7179 C.N.R.S/M.N.H.N, Museum National d'Histoire Naturelle, Paris, France
[3] Department of Biology, Washington University, St Louis, MO, USA

## ABSTRACT

Invasive species are a global threat to biodiversity. Cases where the invasion has been tracked since its beginning are rare, however, such that the first interactions between invasive and native species remain poorly understood. Communication behavior is an integral part of species identity and is subject to selection. Consequently, resource use and direct interference competition between native and invasive species may drive its evolution. Here, we tested the role of interactions between the recently introduced invasive lizard *Anolis cristatellus* and the native *Anolis oculatus* on variation in behavior and communication in Calibishie (Dominica). From May to June 2016, we filmed 122 adult males of both species displaying in banana farms under two contexts (allopatry and sympatry). We then recorded (i) the proportion of time spent displaying and (ii) the relative frequency of dewlap vs. push-up displays. To control for habitat variation, we measured and compared the habitat characteristics (canopy openness and habitat openness) of 228 males in allopatry and sympatry. While the habitat characteristics and total display-time did not differ between the contexts for the two species, the proportion of display-time spent dewlapping by *A. cristatellus* decreased in sympatry. The display of *A. oculatus* did not differ between the contexts, however. Shifts in microhabitat use, predation pressure, or interspecific interference are potential factors which might explain the behavioral changes in display observed in *A. cristatellus*. This study highlights the role of behavioral traits as a first response of an invasive species to recent competition with a closely related native species.

## INTRODUCTION

Invasive species are a global threat to biodiversity, driving species to extinction and imperiling ecosystems (*Parmesan, 2006*; *Van der Putten, 2012*). Therefore, understanding how invasive species successfully establish in new environments and their impacts on native species have become some of the main contemporary challenges. However, only rarely are invasions tracked from their beginning. Yet, the first years of native–invasive

Corresponding author
Claire M. S. Dufour,
clairems.dufour@gmail.com

species competition often determine its outcome (*Puth & Post, 2005*). By consequence, recent species invasions constitute an important field of research in evolutionary conservation biology by providing a natural experimental setting to test the role of interspecific competition on species evolution in action.

Evolutionary biologists have often considered behavior as an inhibitor of evolutionary change (*Bogert, 1949*), allowing individuals to avoid selection imposed by novel ecological contexts (reviewed in *Huey, Hertz & Sinervo, 2003*; *Duckworth, 2009*; *Muñoz & Losos, 2018*). However, behavioral changes may directly alter selective pressures (*Mayr, 1963*; *Duckworth, 2009*), insofar as they modify the interaction between individuals and their environment by determining how organisms forage (*Grant & Grant, 2014*), avoid predators (*Losos, Schoener & Spiller, 2004*), mate (*Lande, 1981*), maintain homeostasis (*Muñoz & Losos, 2018*), and respond to competitors (*Anderson & Grether, 2010*). From this perspective, while ecologists have focused on genetic, ecological and life-history characteristic of invasive species, the behavioral mechanisms determining the outcome of species competition deserve more attention (*Holway & Suarez, 1999*; *Mooney & Cleland, 2001*). For example, native Californian ants were displaced by the invasive Argentine ant (*Lepithema humile*) due to behavioral adaptations of the invasive species (*Holway, 1999*; *Holway & Suarez, 1999*; *Human & Gordon, 1999*).

Communication is subject to natural and sexual selection and is at the forefront of species divergence and recognition processes (*Ord, Stamps & Losos, 2010*; *Macedonia et al., 2013*; *Wong & Candolin, 2015*). Exploitative (resource use; *Huber & Podos, 2006*; *Huber et al., 2007*), direct interference (*Anderson & Grether, 2010*), and reproductive (*Höbel & Gerhardt, 2003*) competition between closely related species may drive its evolution. As a result, communication and display behavior are particularly likely to evolve in the context of interactions between native and invasive species. Nonetheless, despite the potential of invasive species to exert selection on native signalers (*Servedio, 2004*), the role of native–invasive species competition in the evolution of communication behavior remains poorly studied (*Candolin & Wong, 2012*).

The present study aims to examine changes in communication and display behavior during the first stages of an invasion. Specifically, we studied interspecific interactions between an invasive species, *Anolis cristatellus*, from Puerto Rico and the native *Anolis oculatus* on the island of Dominica. The introduction history of *A. cristatellus* has been well documented in Dominica as this species was inadvertently introduced in 1998–2000 on the south Caribbean coast (*Eales, Thorpe & Malhotra, 2008*, *2010*). Since then, *A. cristatellus* arrived in Calibishie in the North-eastern region no earlier than 2014. The species have been shown to fight with each other and diverge in their microhabitat use (i.e., perch height) in sympatry (*Dufour, Herrel & Losos, 2017*). In addition, because the spread of *A. cristatellus* has been patchy (due to the random spread of this species along the main road), allopatric populations occur in Calibishie, allowing the comparison of behavioral and ecological traits in the two contexts (i.e., allopatry vs. sympatry) for the two species.

Lizards of the genus *Anolis* have a colorful and retractable throat fan (dewlap) used to attract females, and repel rivals and predators (*Jenssen, 1977*; reviewed in *Losos, 2009*).

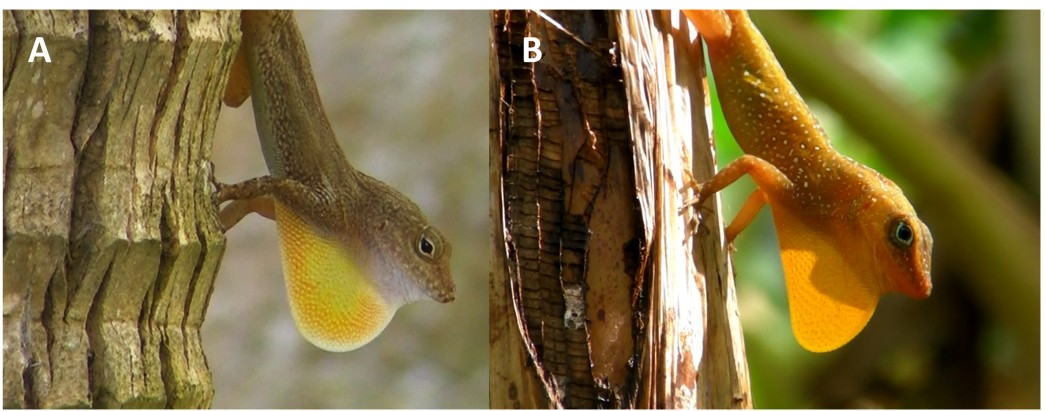

**Figure 1** *Anolis cristatellus* (A) and *A. oculatus* (B) males displaying (dewlap extension) in Calibishie (**Dominica, 2016**). Photo source credit: CMS Dufour.

Display behavior (mostly observed in males) is composed of a specific sequence of dewlap extensions and push-ups (Fig. 1). Both push-up display and dewlap extensions may be used as long distant signals (*Losos, 1985*; *Fleishman, 1992*; *Irschick & Losos, 1996*; *Ord & Stamps, 2008*) but their relative functions are not well-known. Nonetheless, while the vertical body movements appear to be equally important for fights in all anoles (*Lailvaux & Irschick, 2007*), the dewlap extension display seems to be more frequent in more territorial species (*Hicks & Trivers, 1983*; *Losos, 1990*; *Irschick & Losos, 1996*).

Movement of the background vegetation (*Ord et al., 2007*; *Ord & Stamps, 2008*), predation (*Leal & Rodríguez-Robles, 1997*), and species recognition (*Ord & Martins, 2006*; *Macedonia et al., 2015*) are all important drivers that shape anole display behavior and which might be impacted by interspecific competition. For instance, the commonly observed perch use divergence resulting from interspecific competition in *Anolis* lizards (*Williams, 1972*, *1983*; *Stuart et al., 2014*; *Dufour, Herrel & Losos, 2017*) might induce new microhabitat pressures in terms of predation, light or vegetation movement. In addition, interspecific interference (*Grether et al., 2013*) and reproductive competition (*Ord & Martins, 2006*) might shape the display in anoles (but see *Hess & Losos, 1991*).

From May to June 2016, we filmed male *A. oculatus* and *A. cristatellus* displaying in the field and recorded (i) the proportion of time spent displaying and (ii) the relative frequency of dewlap vs. push-up displays. To test the effect of interspecific competition on the measured traits, we took advantage of the fact that allopatric and sympatric populations of the two species live in similar environments (banana farms) within the same climatic and altitudinal region. We also tested whether the general habitat characteristics (i.e., canopy openness and habitat openness) were similar in allopatry and sympatry. If display behavior is one of the first responses to recent interspecific competition, its duration, characteristics, or both should differ in sympatry compared to allopatry, assuming that habitats are similar. Alternatively, differences in habitat characteristics between allopatric vs. sympatric populations may lead to differences in display behavior independently of effects of interspecific competition.

## MATERIAL AND METHODS

This study was performed under the research permit from the Ministry of Agriculture and Fisheries, Forestry, Wildlife, and Parks division of Dominica and with all the IACUC (n° 26-11) authorizations from Harvard University.

### Study sites and species

From May 1st to June 9th 2016, we sampled four sites at which both species occurred ("sympatric"), two sites at which only *A. cristatellus* occurred and three sites at which only *A. oculatus* occurred (the latter two sites termed "allopatric") within the Calibishie region in Dominica (Fig. 2). The allopatric populations of the invasive species may be the result of the extinction of the native species. Nonetheless, the recent arrival of *A. cristatellus* in Calibishie–2014—and the fact that we recorded extremely low population densities of *A. oculatus* (and no *A. cristatellus*) in some banana farms suggest that the allopatric populations of the invasive species result from its establishment in naturally unoccupied banana farms. To minimize the influence of the habitat characteristics on display behavior, populations were sampled in banana farms. Each site was sampled on three to five consecutive days. To prevent the risk of re-sampling the same individual within a field session, lizards were captured by noose or hand and marked with a non-toxic marker after recording and filming. Each sampled individual was replaced at the exact same spot within 10 h after capture.

### Display behavior

A total of 122 adult males, observed for the first time in a sitting position (*A. cristatellus* in allopatry ($n = 23$) and sympatry ($n = 30$); *A. oculatus* in allopatry ($n = 31$) and sympatry ($n = 38$)) were video recorded directly in the field. To record undisturbed behavior, the camera was positioned perpendicular to the long axis of the focal lizard in the horizontal plane and at a distance of at least five meters. Recording started when the focal individual initiated the first display. Recording sessions (mean ± SD: 8.24 ± 3.20 min) were long enough to observe several displays while maximizing the number of tested individuals (the recording stopped when the lizard moved away). With the software JWatcher, (i) the proportion of time spent displaying and (ii) the proportion of display-time spent dewlapping vs. push-ups were recorded by the same observer (all displays were categorized as either dewlap or push-up displays; our metric was the proportion of display-time spent in dewlap displays, which is a measure of the relative time spent in the two types of displays).

### Habitat characteristics

The habitat characteristics of a total of 81 *A. cristatellus* and 147 *A. oculatus* adult males were determined in allopatry and sympatry by measuring the canopy openness (as the number of squares with more than 50% of visible sky, measured with a Ben Meadows spherical densiometer, convex model) and the habitat openness (distance in cm to the closest perch available at the same horizontal plan than where the focal lizard was spotted) from the perch where the lizard was initially observed.

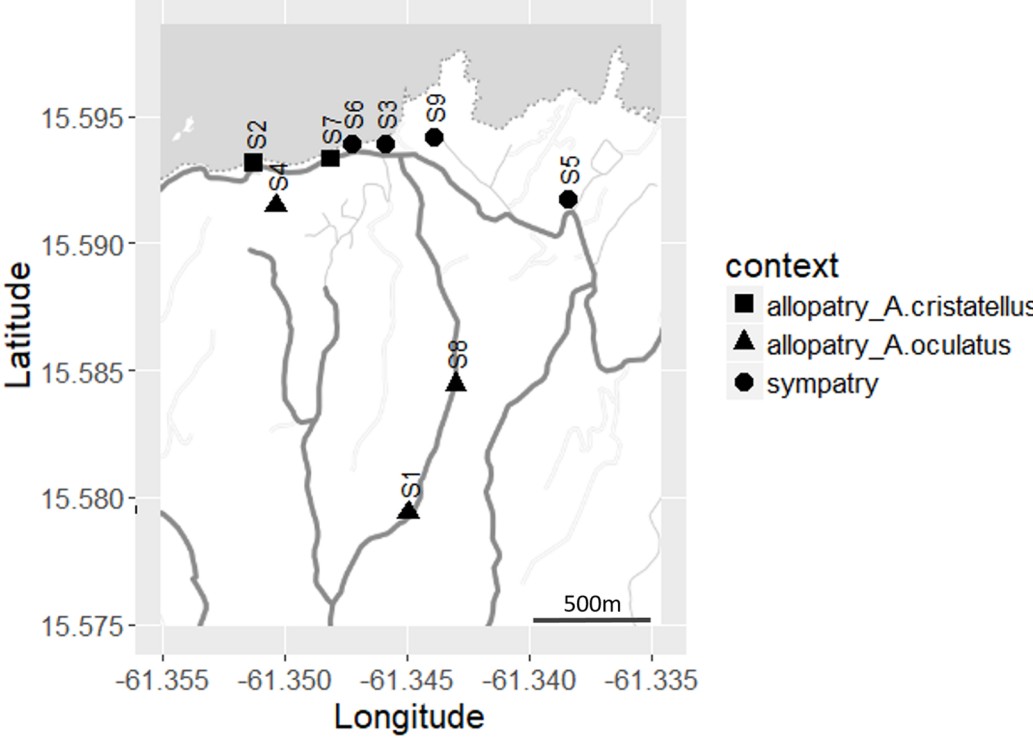

**Figure 2 Distribution of the sites sampled (S1–S9) across Calibishie (Dominica, 2016).** Different shapes indicate context (allopatry, sympatry) and species sampled. The grey and white areas represent the sea and land, respectively. The country border (dashed line), the roads (thick grey lines) and the unpaved paths (thin grey lines) are represented. Map credit: Stamen Design, under a Creative Commons Attribution (CC BY 3.0) license.              

## Statistical analyses

Statistical analysis was conducted with R-v3. (*R Development Core team, 2011*). Normality and heteroscedasticity of distributions were verified graphically (data were log-transformed when necessary). The proportions of (i) total displaying and (ii) relative frequency of dewlap vs. push-up displays were analyzed with linear mixed-effect models, testing for the effect of context (allopatry vs. sympatry), species and the interaction between the two as factors and site as random effect. Linear mixed-effect models were performed on the log-transformed canopy openness and habitat openness data, with context as factor and site as random effect. When a two-way interaction was significant, post hoc analyses (Tukey test) were performed by separating the two species and testing the effect of context.

## RESULTS

The proportion of time spent displaying did not differ significantly between the two contexts (Table 1; Fig. 3).

The proportion of display-time spent dewlapping vs. performing push-up displays was significantly lower in sympatry compared to allopatry for *A. cristatellus* (Tukey, d.f. = 7,

**Table 1 Statistical results from the final linear mixed-effect models (based on AIC) testing the behavioral traits and ecological characteristics of *Anolis cristatellus* and *A. oculatus* adult males according to the variables (i.e. species, context and the interaction of the two). The site was set as a random effect.**

|  | Trait | Variable | Value | SE | d.f. | *t*-value | *P*-value |
|---|---|---|---|---|---|---|---|
| Behavioral traits | Proportion of total display | Intercept | 0.062 | 0.017 | 112 | 3.746 | <0.001 |
|  |  | Context | −0.011 | 0.019 | 7 | −0.553 | 0.597 |
|  |  | Species | 0.033 | 0.016 | 112 | 2.064 | 0.041 |
|  | Proportion of display-time spent dewlapping | Intercept | 0.636 | 0.054 | 111 | 11.708 | <0.001 |
|  |  | Context | −0.225 | 0.072 | 7 | −3.121 | 0.017 |
|  |  | Species | 0.277 | 0.072 | 111 | 3.869 | <0.001 |
|  |  | Context: species | 0.205 | 0.096 | 111 | 2.138 | 0.034 |
| Ecological characteristics | Habitat openness | Intercept | 3.313 | 0.095 | 361 | 35.052 | <0.001 |
|  |  | Context | 0.147 | 0.104 | 7 | 1.416 | 0.199 |
|  |  | Species | 0.234 | 0.097 | 361 | 2.418 | 0.016 |
|  | Habitat canopy cover | Intercept | 1.469 | 0.157 | 361 | 9.352 | <0.001 |
|  |  | Context | 0.203 | 0.189 | 7 | 1.076 | 0.318 |
|  |  | Species | −0.153 | 0.131 | 361 | −1.163 | 0.246 |

$t = 3.121$, $P = 0.016$), but did not change for *A. oculatus* (Tukey, d.f. = 7, $t = 0.323$, $P = 0.756$, Fig. 4; Table 1).

Canopy openness (Fig. 5) and habitat openness (Fig. 6) did not differ significantly in allopatry and sympatry for the two species (Table 1).

## DISCUSSION

Invasive species are a global scourge, but data on the interactions between native and invasive species when they first come into contact are rare (*Puth & Post, 2005*). Our study revealed that, only two years after their arrival in Calibishie, males of *A. cristatellus* showed a shift in the type of displays performed in sympatry compared to allopatry, performing relatively more push-ups and fewer dewlap displays. No change was observed for the native *A. oculatus*. The similarity of the habitat characteristics (i.e., canopy openness and habitat openness) between the two contexts suggests an important role for recent competition in driving the behavioral change observed in the invasive species. The following discussion addresses the potential role of predation, microhabitat use, and agonistic interaction as possible explanations for the differences between species in their response to sympatry.

Microhabitat use has been shown to be an important driver shaping communication behavior in species in general, and in *Anolis* lizards in particular (*Ord, Stamps & Losos, 2010*). For instance, the movement of the visual background and predation pressure are among the main factors driving communication behavior in anoles and in *A. cristatellus* in particular (*Leal & Rodríguez-Robles, 1995*, *1997*; *Ord et al., 2007*). Moreover, the role of interspecific competition in microhabitat species divergence has been demonstrated in anoles (*Schoener, 1970*; *Williams, 1972*, *1983*; *Losos, 2009*; *Stuart et al., 2014*).

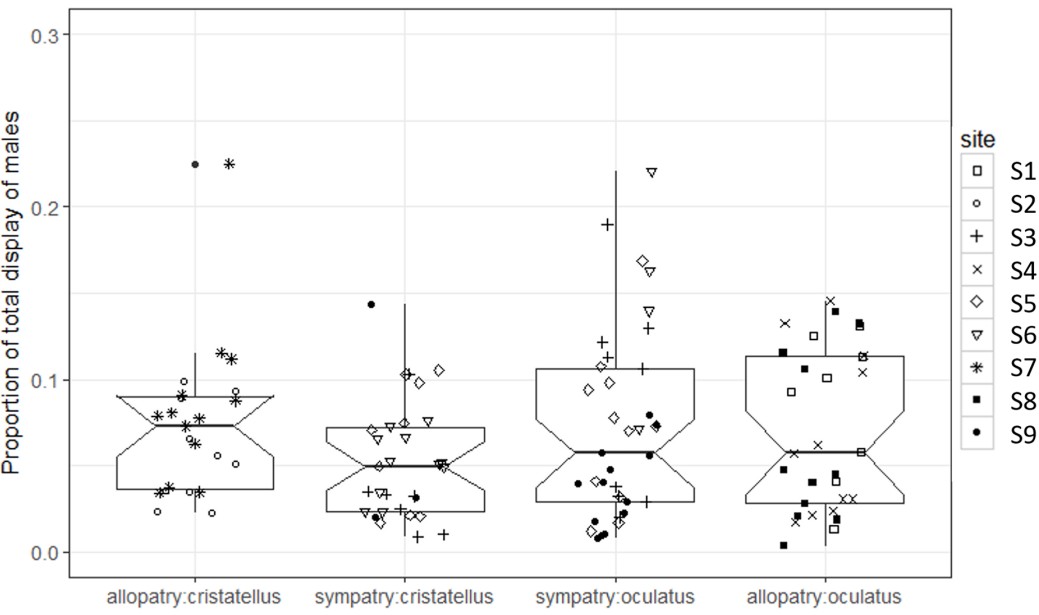

**Figure 3 Proportion of time spent displaying by male *Anolis* recorded in Calibishie (Dominica, 2016) across species (*A. cristatellus, A. oculatus*) and context (allopatry, sympatry) and according to the sites of sampling.** Box-plots (calculated from all individuals) show the median (thick line), first and third quartiles. The lines extending vertically from the boxes indicate the lowest datum still within 1.5 IQR (interquartile range) of the lower quartile, and the highest datum still within 1.5 IGR of the upper quartile. Individual points beyond these lines represent outliers. The notches indicate 95% confidence intervals so that the distributions differ significantly if the notches do not overlap.

Nonetheless, studies on the direct link between recent habitat character displacement and the evolution of communication behavior are lacking and no such studies have been published on anoles.

In Dominica, and in Calibishie in particular, *A. cristatellus* and *A. oculatus* diverged in sympatry in perch height: the invasive species moved downward toward the ground while the native species used higher perches compared to populations in allopatry (*Dufour, Herrel & Losos, 2017*). This microhabitat divergence might be correlated with a different visual background, potentially driving display variation. Indeed, the visual background is expected to be more variable higher up (due to foliage motion) than on the ground. It has been shown that the duration and the speed of the display of *Anolis* lizards increased in habitats with greater movement of the background (*Ord et al., 2007*; *Ord, Stamps & Losos, 2010*). Moreover, the Australian lizard *Amphibolurus muricatus* changed the structure of its communication behavior and increased the duration of its tail display in a habitat characterized by background movement (*Peters, Hemmi & Zeil, 2007*). In our study, a more stable background lower to the ground may be associated with the shift toward displaying more with push-ups and less with dewlap displays observed in *A. cristatellus* in sympatry. Indeed, the time and energetic costs of the dewlap extension display may induce a trade-off between conspicuousness and metabolic cost (*Vehrencamp, Bradbury & Gibson, 1989*; *Marler et al., 1995*; *Clark, 2012*).

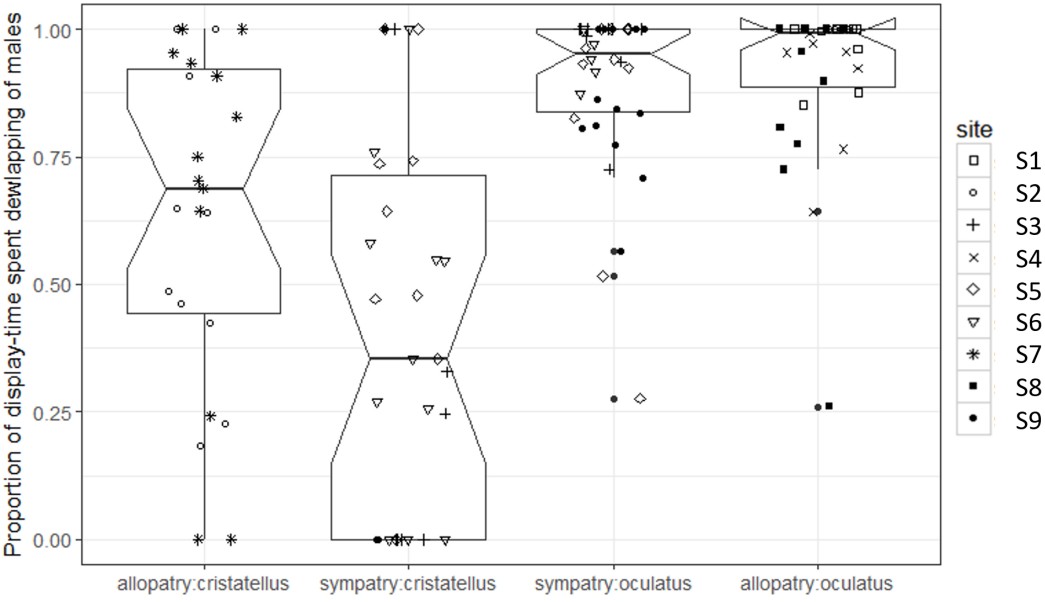

**Figure 4 Proportion of display-time spent dewlapping versus performing push-up displays by male *Anolis* video-recorded in Calibishie (Dominica, 2016) across species (*A. cristatellus*, *A. oculatus*) and context (allopatry, sympatry) and according to the sampling sites.** Box-plots (calculated from all individuals) show the median (thick line), first and third quartiles. The lines extending vertically from the boxes indicate the lowest datum still within 1.5 IQR (interquartile range) of the lower quartile, and the highest datum still within 1.5 IQR of the upper quartile. Individual points beyond these lines represent the outliers. The notches indicate 95% confidence intervals so that the distributions differ significantly if the notches do not overlap.

Why the inverse pattern is not observed in *A. oculatus*, which perches higher in sympatry, is unclear, but could be related to a less drastic difference of the movement of the background vegetation between the two contexts for this species as it is always perching relatively high in trees.

The ecological character displacement in microhabitat use may also induce differences in predation pressure between the two contexts. Indeed, terrestrial anole predators such as rats or *Ameiva* lizards were found in abundance at the study sites. *A. cristatellus* performs a push-up display in presence of a snake predator and increases the rate thereof when the predator is closer (*Leal & Rodríguez-Robles, 1997*). Moreover, the dewlap is a colorful visual signal (*Losos, 1985*; *Leal & Fleishman, 2004*; *Nicholson, Harmon & Losos, 2007*; *Ng et al., 2013*; *Ingram et al., 2016*) and conspicuousness has been shown to increase predation rate in lizards (*Fitch & Henderson, 1987*; *Stuart-Fox et al., 2003*; *Husak et al., 2006*). By consequence, the increase of the push-up display proportion of *A. cristatellus* in sympatry perching lower to the ground may be the result of evolutionary trade-off between predation and communication (*Steinberg et al., 2014*).

Alternatively, direct agonistic encounters between the two species might drive the display behavior shift observed in *A. cristatellus* in sympatry. Indeed, we observed the native *A. oculatus* initiating interspecific agonistic encounters, forcing *A. cristatellus* to move downward. Moreover, *A. oculatus* has a bigger head and can bite harder than
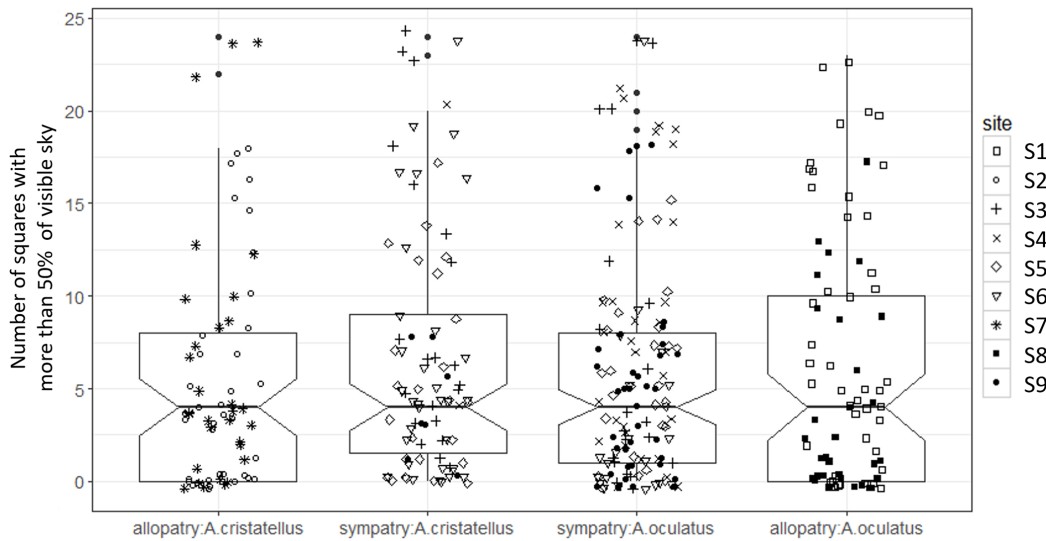

**Figure 5 Canopy openness (number of densiometer squares out of 24 with more than 50% of visible sky) of the habitat used by adult male *Anolis* from Calibishie (Dominica, 2016) across the species (*A. cristatellus*, *A. oculatus*) and the context (allopatry, sympatry) and according to the sites of sampling.** Box-plots (calculated from all individuals) show the median (thick line), first and third quartiles. The lines extending vertically from the boxes indicate the lowest datum still within 1.5 IQR (interquartile range) of the lower quartile, and the highest datum still within 1.5 IQR of the upper quartile. Individual points beyond these lines represent the outliers. The notches indicate 95% confidence intervals so that the distributions differ significantly if the notches do not overlap.

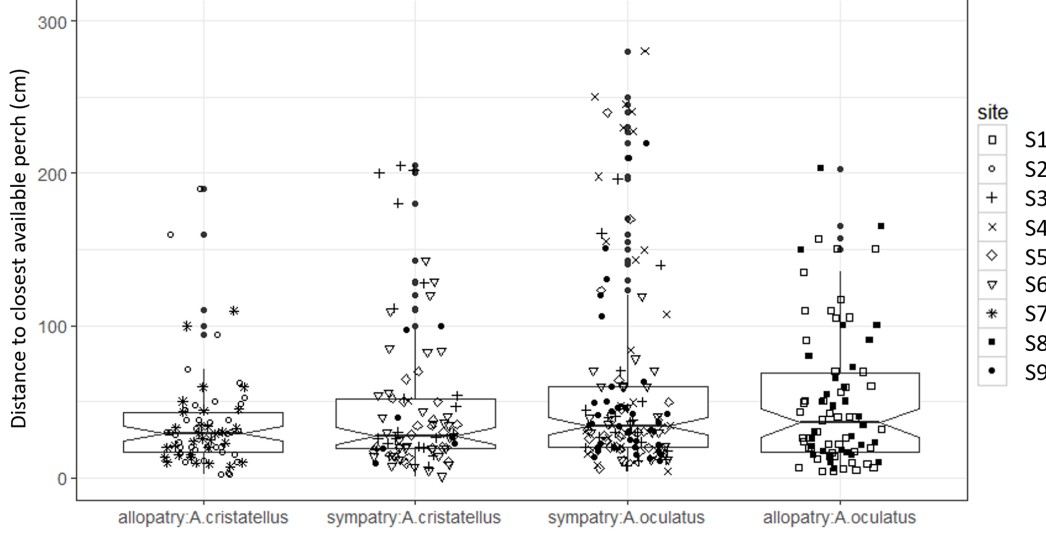

**Figure 6 Habitat openness (distance to closest available perch, cm) of the habitat used by adult male *Anolis* from Calibishie (Dominica, 2016) across the species (*A. cristatellus*, *A. oculatus*) and the context (allopatry, sympatry) and according to the sites of sampling.** Box-plots (calculated from all individuals) show the median (thick line), first and third quartiles. The lines extending vertically from the boxes indicate the lowest datum still within 1.5 IQR (interquartile range) of the lower quartile, and the highest datum still within 1.5 IQR of the upper quartile. Individual points beyond these lines represent the outliers. The notches indicate 95% confidence intervals so that the distributions differ significantly if the notches do not overlap.

*A. cristatellus*, suggesting the dominant status of the native species during interspecific fights (C. M. S. Dufour, J. B. Losos & A. Herrel, 2016, unpublished data). Thus, by decreasing dewlap extension time, *A. cristatellus* might be more cryptic in sympatry to avoid the agonistic encounters with *A. oculatus*.

It is possible that the two types of display observed in *A. cristatellus* might be the result of different social contexts (assertion, courtship or challenge; *Carpenter, 1967*; *Jenssen, 1977*) encountered in allopatry and sympatry. However, more recent studies have revealed that the characterization of different display types depending to the social context confuses the form and the function of the displays (*Decourcy & Jenssen, 1994*; *Lovern et al., 1999*; *Bloch & Irschick, 2006*).

## CONCLUSION

To conclude, this study reveals the presence of character displacement (*Brown & Wilson, 1956*) in elements of the behavioral display in the invasive species *A. cristatellus* in Dominica. More research is required to elucidate whether these display shifts are plastic or the result of genetic change. As plasticity has been suggested to account for most of the display behavior variation in *Anolis* lizards (*Ord, Stamps & Losos, 2010*), this is also likely the case here. This study represents a rare case in which the impact of competition between native and invasive species is studied at the early stages of the invasion process, highlighting the importance of the communication behavior as one of the first responses to environmental change.

## ACKNOWLEDGEMENTS

We thank Jonathan Suh for his assistance on the field and the reviewers for their valuable comments.

### Funding
The field work was supported by the Putnam grant and Fyssen fellowship. The funders had no role in study design, data collection and analysis, decision to publish, or preparation of the manuscript.

### Grant Disclosures
The following grant information was disclosed by the authors:
Putnam grant.
Fyssen fellowship.

### Competing Interests
The authors declare that they have no competing interests.

### Author Contributions
- Claire M. S. Dufour conceived and designed the experiments, performed the experiments, analyzed the data, contributed reagents/materials/analysis tools, prepared figures and/or tables, authored or reviewed drafts of the paper, approved the final draft.

- Anthony Herrel conceived and designed the experiments, contributed reagents/ materials/analysis tools, authored or reviewed drafts of the paper, approved the final draft.
- Jonathan B. Losos conceived and designed the experiments, contributed reagents/ materials/analysis tools, authored or reviewed drafts of the paper, approved the final draft.

### Animal Ethics

The following information was supplied relating to ethical approvals (i.e., approving body and any reference numbers):

This study was performed with authorization from the Harvard University IACUC (protocol # 26-11).

### Field Study Permissions

The following information was supplied relating to field study approvals (i.e., approving body and any reference numbers):

This study was performed under a research permit from the Ministry of Agriculture and Fisheries, Forestry, Wildlife and Parks division of Dominica.

### Data Availability

The raw data are provided in the Supplemental Tables.

### Supplemental Information

Supplemental information for this article can be found online at http://dx.doi.org/ 10.7717/peerj.4888#supplemental-information.

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
