# Peer review of "The effect of recent competition between the native Anolis oculatus and the invasive A. cristatellus on display behavior"

_PeerJ, doi:10.7717/peerj.4888_

## Round 0.1 · original submission · Major Revisions

Overview

Both reviewers found your study interesting and generally well researched and well presented. I agree with this view. However, there are some issues to be considered. Because one of the concerns is the statistical analysis, I have categorized the need as 'major revision'. You can consider my editor's comments below as if they were a third review: make changes if the comments are valid and provide a detailed rebuttal if they are not.

Editor's Comments

1) Replication. Your study borders on pseudoreplication. You have treated individual courtship records as the units in your analysis, but the small number of sites means that site differences might also explain the pattern. I am sensitive to the challenges of replication in field studies of habitat variation as I have been criticized for lack of replication of reserves in studies of the only marine reserve in Barbados. Nevertheless, it is important to take the possible effect of site, independent of sympatry, into account. I suggest that you undertake an analysis that includes site as a factor to at least partially address this problem. In addition, it would be appropriate to revise Figures 2 and 3 to show the data for each site separately. Finally, your Discussion should acknowledge the potential for a site effect and discuss its possible influence.

2) Other issues with Figures 2 and 3. If the data are not normally distributed, requiring a non-parametric analysis, parametric descriptive statistics (mean, SE) are not appropriate. Consider using non-parametric box plots. I am not sure that a strip chart, recommended by Reviewer 2 would be the best way to do this, but consider alternative ways to display the data most clearly. Also, in addition to the use of color, distinguish the species by graphical means that will be clear to readers who might not have a color copy.

3) Site location. I agree with Reviewer 2 that you should identify the location of your sites, perhaps providing a map with scale and latitude and longitude indicators and in the Methods include distances. It is also not clear to me how you can have allopatric populations of the invasive species if they invade the habitat of the resident species. Would a bit more explanation of the occurrence of allopatric and sympatric sites be worthwhile?

4) Literature on effect of invasive species on communication behavior. I agree with Reviewer 1 that you should briefly review the previous literature on effects of species invasion on communication. This is needed to emphasize the contribution of the study and to support the final statement of your conclusion (L190-193)

Minor issues:

L1,16. Title and Abstract. 'The role of competition on display' is not correct word use. 'The effect of competition on display' would be more usual. 'Role' has more of a connotation of 'function' and usually is accompanied by 'in', e.g., 'the role of X in process Y'. I suggest replacing 'role' by 'effect'. If you are trying to avoid the causal implication of 'effect' as your study is not experimental, some other wording might be more appropriate.

L25 (and many other places). Microhabitat is usually one word, not hyphenated.

L100. 'perpendicularly' does not adequately describe the camera position because it does not state in reference to what plane. It could be perpendicular above or below or in front of the animal. Do you mean perpendicular to the long axis of the anole?

L119ff. I don't think you need sub-heads to divide the very short Results section. This would be true even if the section is expanded in keeping with some of the review suggestions.

L128. Include the values for habitats as well as the statistical comparisons, whether in the text or in a figure as suggested by a reviewer.

L140. Microhabitat use (no hyphen)

L167. 'predators such as rats'

L168. 'search for prey'

L169. 'and increases the rate when the predator is closer'

L172. 'conspicuousness has been shown'

L190. Comma after '2010)'

L193. I don't understand your claim that your study highlights the importance of communication behavior as one of the first responses to environmental change. The change in communication occurred in the invasive species not the native species; is environmental change the most relevant context for a species invading a new area?

L198ff. Please check all your references very carefully. Many species names are not in italics, some journal titles are in all caps, some journal articles lack page numbers, some journal titles are not fully capitalized, some standard details are missing from books and edited volumes.

Fig. 1. Because you generally presented information on cristatellus first, it might be best to present the image first in the caption and on the left panel to help the reader keep the two species straight.

Reviewer 1 ·

Basic reporting

The authors investigated whether human-mediated introduction of Anolis cristatellus to the distribution area of native Anolis oculatus changed communication behavior of these two closely related species in wild. The result clearly showed that invasive A. cristatellus changed display behavior in sympatry with the congener compared to that in allopatry, whereas A. oculatus did not change its display between sympatric and allopatric population. Although the data did not fully demonstrate the change in behavior was caused by interspecific interaction, this provide a rare example of early response of an invasive species to recent secondary contact with a closely related native species.

Experimental design

Although this is a well-written paper that presents interesting data in general, I believe authors can analyze the data more; to consider whether the difference of display between contexts was induced by interspecific interaction, correlation between degree of habitat displacement and behavioral displacement in each site should be also checked. These displacements would positively correlate if both change were caused by interspecific interaction. Comparison of body size and BMI may be also informative to evaluate presence of interspecific competition. In addition, boxplots showing habitat characteristics would be useful for readers to know how different average and variation of habitat characteristics between species and context.

Validity of the findings

All in all, the findings drawn from the data presented was enough valid. The only concern is it was unclear whether social function of dewlap extension and push-up is fully same. This information would be important to interpret the change of the proportion of time spent displaying.

Additional comments

It will be of interest to readers of this journal, PeerJ, particularly researchers in evolutionary ecology and behavioral ecology. However, in the Discussion section, the authors mostly stated about the results and previous researches on anoles. I think the authors should refer to other studies about behavioral response of introduced or native organisms when they contact secondarily.

Reviewer 2 ·

Basic reporting

In this manuscript, the authors compare display behaviors of a native and introduced anole species at different sites in Calibishie, Dominica to identify if display characteristics change when the species are in allopatry versus sympatry. The authors have a great system to study this question with the introduction history of the invasive species well known and recent. The manuscript is well-written and an interesting read.

Experimental design

I suggest that the research question is more hypothesis driven. After the introduction, I was left wondering why the display differ with increased competition, especially if the two species have diverged in microhabitat? The authors offer some hypotheses in the discussion, but I think the introduction could be made stronger if there was more of a prediction to help lead the reader along.

Validity of the findings

I am curious why the authors decided to focus on the dewlap extensions and not the push up behavior. Push ups are actually thought to be more conspicuous and used for long distance communication compared to dewlap extensions (e.g. Fleishman 1992, Ord and Stamps 2008) – the opposite of what the authors speculate (lines 175-177) – so if A. cristatellus increased push up time in sympatry, could this be a sign of more aggressive behavior? Some discussion of the push up part of the display would strengthen the paper.

This paper could also be strengthened by some acknowledgement that anoles use different types of displays (Type A, Type B; Jenssen 1977) presumably in different contexts (e.g. “challenge” versus “assertion” displays, Carpenter 1967). It seems that the authors would have captured a variety of different displays in different societal contexts - could the differences between sympatric and allopatric sites be due to the anoles using one type of display more often than another?

I also would like to see Figures 2 & 3 as a strip chart so all the data points are seen. It would be helpful to see how disparate the displays really are among the different sites, rather than just the mean and standard error. Along these lines, I would have liked a better sense about the different sites. For example, how far apart are the different sites? Some detail in the text or, even better, a map of the sites would be useful. Also, how much variation is there among the different sites? If the authors present Fig. 2 & 3 as a strip chart, they could even show the data points from different sites by representing the data points as different symbols.

---

## Round 0.2 · accepted · Accept

The changes are satisfactory, and the manuscript is now suitable for publication. A number of minor changes in wording are indicated on the annotated pdf. I have discussed all changes other than minor grammatical issues with the author. The author has provided an updated Word document with new figure captions and revised Table 1. I had to make additional corrections on this document, in consultation with the author, and will send this to PeerJ staff.

#